# Agricultural Innovations to Reduce the Health Impacts of Dams

Andrea J. Lund [1],*, David Lopez-Carr [2], Susanne H. Sokolow [3,4], Jason R. Rohr [5] and Giulio A. De Leo [6]

1 Emmett Interdisciplinary Program in Environment and Resources, Stanford University, Stanford, CA 94305, USA
2 Department of Geography, University of California, Santa Barbara, CA 93106, USA; davidlopezcarr@ucsb.edu
3 Woods Institute for the Environment, Stanford University, Stanford, CA 94305, USA; shsokolow@gmail.com
4 Marine Science Institute, University of California, Santa Barbara, CA 93106, USA
5 Department of Biological Sciences, University of Notre Dame, Notre Dame, IN 46556, USA; jrohr2@nd.edu
6 Hopkins Marine Station, Stanford University, Pacific Grove, CA 93950, USA; deleo@stanford.edu
* Correspondence: andrea.janelle.lund@gmail.com

**Abstract:** Dams enable the production of food and renewable energy, making them a crucial tool for both economic development and climate change adaptation in low- and middle-income countries. However, dams may also disrupt traditional livelihood systems and increase the transmission of vector- and water-borne pathogens. These livelihood and health impacts diminish the benefits of dams to rural populations dependent on rivers, as hydrological and ecological alterations change flood regimes, reduce nutrient transport and lead to the loss of biodiversity. We propose four agricultural innovations for promoting equity, health, sustainable development, and climate resilience in dammed watersheds: (1) restoring migratory aquatic species, (2) removing submerged vegetation and transforming it into an agricultural resource, (3) restoring environmental flows and (4) integrating agriculture and aquaculture. As investment in dams accelerates in low- and middle-income countries, appropriately addressing their livelihood and health impacts can improve the sustainability of modern agriculture and economic development in a changing climate.

**Keywords:** dams; agriculture; livelihoods; health; schistosomiasis; restoration; sustainable development; climate adaptation





## 1. Introduction

As infrastructure that enables production of both food and energy, dams are a key component of climate change adaptation [1]. They are meant to improve agricultural productivity and reduce vulnerability to droughts [2]. Especially in semi-arid regions experiencing desertification, water management infrastructure for food and energy production is necessary to sustain a growing human population and promote economic development in a changing climate. Yet, dams face many criticisms, ranging from cost overruns [3] and population displacement [4] to ecosystem disruption [5] and increased transmission of infectious disease [6]. Addressing these shortcomings—namely, the uneven distribution of dams' costs and benefits and the health impacts of dams driven by their environmental disruption—is crucial to maximize the social return on infrastructure investment and ensure the sustainability of modern agriculture in dammed landscapes.

Dams have displaced an estimated 40–80 million people globally in the last 50 years [4]. Dams designed to generate hydropower often transmit electricity to urban areas without extending the same benefits to rural areas [7,8]. While in some places dams supporting irrigation increase agricultural production and reduce vulnerability to droughts, in other places, they can increase poverty and variability in crop yields [9,10]. These disparities and uncertainties may be driven in part by the investment required to undertake irrigated agriculture compared to traditional practices. Compared to traditional cultivation of recession crops in floodplains, for example, irrigated agriculture requires greater investment in and use of agrochemical inputs such as fertilizers and pesticides, up-front costs that

may be difficult for subsistence farmers to afford [2]. Furthermore, by altering natural flood regimes, dams have also made it difficult or impossible to cultivate recession crops in floodplains [11–14], forcing farmers to invest in inputs for irrigated crops or turn to another economic activity entirely.

A number of infectious diseases are also associated with both dams [6,15] and agricultural activity [16]. In particular, the transmission of schistosome parasites increases with proximity to dams and irrigation infrastructure [17]. Changes in hydrological regimes favor the aquatic snail populations that transmit the parasites, while the promotion of agricultural livelihoods keeps people in contact with fresh surface water [18], the route of exposure to infection. The convergence of these processes increases the likelihood of infection with schistosome parasites in endemic areas. In addition, repeated reinfection after treatment threatens the success of ongoing efforts to control the disease [19,20]. The inflammatory pathology of prolonged and repeated infection can lead to organ failure, cancer, and infertility [21] and disrupt cognitive development in children [22] and labor productivity in adults [23,24]. By putting people in contact with a favorable environment for parasite transmission, the negative health impacts of dams likely diminish the intended benefits of increased agricultural activity [25].

The livelihood and health impacts of dams have their roots in a combination of poverty, land-use change, and the associated environmental impacts. Dams compromise both terrestrial and aquatic biodiversity [26,27]. By stemming natural flood regimes, dams disrupt the reproduction of many fish species, affecting inland fisheries [11]. Disruptions to fisheries can, in turn, affect fishing livelihoods and increase the number of people at risk for nutritional deficiencies [28]. Environmental disruption can also facilitate the transmission of infectious diseases. In addition to the hydrological changes that favor populations of medically important snails, biodiversity loss—particularly the loss of aquatic invertebrates that prey on the snails that transmit schistosome parasites [29]—can lead to unchecked growth in snail populations and corresponding increases in parasite transmission. The transition from traditional to modern agricultural regimes also increases reliance on agrochemicals, as the sediment that provides nutrients for recession agriculture remains impounded behind dam barriers along with flood waters. The use of agrochemicals can exacerbate disease dynamics, because insecticides can cause greater harm to snail predators than to snails and fertilizers can promote the growth of both snail food and vegetative habitat [30,31].

While some of these impacts prompted funders to withdraw from dam projects in previous decades [32], the pace of investment in dam infrastructure is growing again [33]. The increasingly tangible impacts of climate change as well as the development trajectories of low- and middle-income countries make dams a key, but controversial, tool in the toolbox of sustainable development. Given this, addressing the livelihood, health and environmental impacts of dams will become central to improving the sustainability of modern agriculture and economic development in a changing climate. We outline agricultural innovations that can mitigate negative impacts of dams and promote sustainable development.

## 2. Agricultural Innovations to Reduce the Negative Impacts of Dams

### 2.1. Restoring Natural Predators of Parasite-Bearing Snails in Dammed Watersheds

Almost half of the 800 million people living in schistosomiasis endemic regions are estimated to be at risk of schistosomiasis because of the loss of snail predators following dam construction [34]. The disappearance of snail predators upstream of dams is thought to have contributed to the burden of schistosomiasis [29]. Aquatic species, such as the African river prawn (*Macrobrachium vollenhovenii*), are known to consume snails that transmit schistosomes [35], but dams interrupt their ability to migrate downriver to reproduce in brackish, estuarine water. A field experiment in northern Senegal shows that re-introducing *M. vollenhovenii* to river access sites in villages reduced the abundance of infected snails as well as the prevalence of schistosome infections in people [36]. While these data were limited in their geographic extent, the native range of *M. vollenhovenii* and closely related species extends across much of coastal Africa, where 90% of cases of schistosomiasis occur.

This impact extends hundreds of kilometers upstream of the dams themselves, such that restoring the ecological connectivity of African rivers could have a far-reaching impact on disease transmission, aquatic biodiversity, and inland fisheries [34].

Investment in aquaculture of *M. vollenhovenii* on the African continent could play an important role in both disease control and food security [37]. However, with dam barriers in place, aquatic migratory species like *M. vollenhovenii* would need to be continuously introduced upstream of dams. Sustained ecological restoration of these species will require modification to dam infrastructure to accommodate their aquatic migration. Designing prawn and/or fish ladders for existing and future dams could enable the restoration and preservation, respectively, of native aquatic species and the ecosystem services they provide for both disease control and food production [38]. Restoring prawns in the wild could support both the biological control of schistosome transmission as well as food production through the re-establishment of an inland fishery that has been lost to dam construction [39]. The establishment of catch guidelines that allow for the harvest of older, larger prawns for human consumption could ensure that younger, smaller prawns remain in the ecosystem for the sake of biological control [37,40]. Such a management system would allow dams (and their benefits) to remain in place while mitigating their negative impacts on livelihoods, food security and health.

### 2.2. Removing Submerged Aquatic Vegetation and Transforming It into an Agricultural Resource

By retaining water and slowing flows, dams encourage the growth of often-invasive aquatic vegetation [41,42]. Additionally, the agricultural expansion and intensification facilitated by dams likely increases fertilizer use in the landscape, which runs off into water sources and further promotes the growth of aquatic vegetation. Invasive aquatic species threaten biodiversity and productive use of river environments [43] while supporting the snails that transmit schistosomes [44,45]. Aquatic vegetation has been associated with both increased shedding of schistosome parasites from snails [46] and increased prevalence of infection in humans [47]. There is also evidence for a mutualistic relationship between snails that transmit schistosomes and submerged vegetation from the genus *Ceratophyllum* [46]. The removal of aquatic vegetation from natural water bodies [48], reservoirs [49], and irrigation canals [50], thus, might be an important intervention for reducing snail abundance and disease transmission while maintaining agricultural productivity.

A potential challenge with sustaining vegetation removal, however, is that it is a resource- and labor-intensive process that likely suffers from the tragedy of the commons. Without private incentives to remove, process and/or transport vegetation, the common good of disease control through vegetation removal may not be realized [51]. One agricultural innovation that might incentivize sustained vegetation removal is to transform removed vegetation into an agricultural resource. For example, submerged aquatic vegetation could be turned into compost to facilitate crop production or used as livestock feed to increase agricultural profitability [51–53]. In essence, these approaches would help to close the nutrient loop, returning nutrient-rich runoff from agricultural fields that is captured in aquatic plants back to food production [51]. Preliminary findings suggest that transforming submerged aquatic vegetation to compost can increase crop production in the Sahel region of Africa (unpublished data). Additionally, when we have harvested submerged vegetation, we have observed cattle and donkeys readily consuming it.

Another agricultural use of vegetation is as fuel for biodigesters. The breakdown of vegetation, produces both a nutrient-rich amendment for crops while simultaneously offering natural gas that can be used for cooking or for powering generators. By reducing reliance on fertilizers, these approaches might, indirectly, further reduce schistosome transmission [30,31,54,55]. These innovations have great potential for using aquatic vegetation to improve agricultural production and profitability. In turn, they help sustain the public health benefits of vegetation removal. Removal of vegetation should be undertaken carefully, however, as the opening of shorelines might attract human activity and inadvertently increase transmission [44,45,56]. We encourage researchers to quantify both the public

health and private agricultural benefits of harvesting snail-inhabited vegetation and to test whether privatization helps to sustain the public health benefits.

### 2.3. Restoring Environmental Flows

In addition to the 40–80 million people directly displaced by dams, almost 500 million people globally live downstream of dams and have been affected by hydrologic alteration [8]. Flow regulation leading to the loss of natural seasonal flooding in dammed watersheds is associated with corresponding losses in inland fisheries, flood recession agriculture and dry season grazing [11]. Such traditional practices form the basis of subsistence livelihoods and food security among river-dependent populations [14,57], losses of which have been estimated to outweigh the benefits of dam construction in some settings [42,58]. The loss of seasonal floods also likely contributes to the increased risk of schistosomiasis in dammed landscapes. The destabilizing effects of seasonal flooding can stem the invasion of aquatic vegetation and keep snail populations from becoming perennially established [6].

Controlled dam releases can be used to restore and maintain ecological processes and natural resources [7]. Dam operations that restore environmental flows can accommodate traditional fishing and cultivation practices that rely on natural flood regimes without imposing substantial trade-offs on the conventional agricultural and energy production purposes of dams [13,59–61]. Particularly for recession agriculture, controlled releases can reduce sediment retained by dam barriers, which provides nutrients for cultivation in the floodplain [62]. These controlled releases, in turn, reduce the need for chemical fertilizers, whose use can support medically important snail populations and, by extension, schistosome transmission [31]. Some studies indicate that modifying dam operations in these ways could reduce schistosome transmission [63,64]. Similar studies of dam management for malaria control suggest that this can be accomplished with few trade-offs in irrigation or electricity generation [65,66].

Investigating how environmental and social benefits can be recovered by adjusting dam management could help diversify agricultural practices and improve agricultural outcomes, while mitigating the negative impacts of dams on livelihoods and health. These factors are seldom explicitly accounted for in decision-making about dam construction and operations [8,67], but they represent a significant opportunity for improving the well-being of the hundreds of millions of river-dependent people affected by dams. The estimated economic benefits of traditional uses of natural floodplains are substantial, even exceeding those of irrigated agriculture [42]. Reductions in the often-unquantified negative health impacts could add to this margin.

### 2.4. Integrating Agriculture and Aquaculture

The benefits of *Macrobrachium* spp. and other aquatic species can be cultivated in aquaculture settings and potentially integrated with irrigation systems [68]. These species can control snail populations within irrigation infrastructure and reduce occupational exposure to schistosomes. Modeling analyses indicate that prawn aquaculture is a profitable undertaking for smallholders that also sustainably controls medically important snails [37]. Especially in the water-intensive cultivation of rice, integrating prawn farming with crop cultivation may be particularly advantageous, efficient, and beneficial [69]. This practice is already common in South Asia [70,71], and could be applied to the rapidly growing area under rice cultivation on the African continent [69]. Identifying and using insecticides that minimize harm to aquaculture species would be necessary for this strategy to succeed [55].

## 3. Discussion

The agricultural innovations we propose represent an opportunity to reconcile the apparent trade-offs between the economic development that motivates dam investment and their unintended consequences. The agricultural landscapes created by dams have been demonized for displacing people, disrupting traditional livelihoods, and making people sick. Especially in areas endemic for environmentally-mediated infectious dis-

eases like schistosomiasis, poverty, food insecurity, malnutrition, and infection can all reinforce each other in disease-driven poverty traps [72]. These dynamics likely diminish the intended benefits of dams. As a result, finding ways to restore dammed watersheds' ecological integrity and making ecologically sound design and management choices in the construction of new dams are important in alleviating poverty, food insecurity, and disease transmission for rural populations that depend on rivers and whose livelihoods and health are negatively impacted by dams. Such reconciliation of trade-offs could therefore promote the resilience of coupled human–natural systems in the face of climate change.

None of the four proposed agricultural innovations requires sophisticated technology, but all will require careful planning and management in addition to the technical capacity and political will to assess and address the hydrologic, environmental and social impacts of dams. The goals and implementation of new management schemes are likely to vary across contexts. The ecological and geographical characteristics of a watershed, the human needs, water uses, structural characteristics and limitations of dam infrastructure will dictate what management strategies are feasible and potentially beneficial for livelihoods, food security and health [7,73]. The Nature Conservancy has set an example in its Sustainable Rivers program [74], undertaking flow restoration projects in the United States in cooperation with the Army Corps of Engineers and documenting 850 projects worldwide that demonstrate how flow restoration can be undertaken at many existing dams as well as designed into new dam construction.

Health and disease control could become an objective in existing and future environmental management and agricultural innovation programs. Design or retrofitting of fish and prawn passages could be considered for all new and existing dams, respectively, within 500 km of coasts. The production and distribution of soil amendments and livestock feeds derived from aquatic vegetation holds promise as a social enterprise, while disease control could become a new criterion in the multi-objective frameworks used to manage dams and their reservoirs. Finally, the common practice of integrating agriculture and aquaculture could be extended to new geographic settings, where they may yield health as well as food security benefits.

By design, these innovations will also require cooperation between disciplines and sectors as well as attention to the needs of communities and the capacity and political will of governments. These requirements are in keeping with the focus of the Sustainable Development Goals (SDGs) on interlinkages between disciplines and sectors, bridging between economic development, health and the environment. The agricultural innovations we have proposed have the potential to improve food security (SDG2), health (SDG3) and equity (SDG10) in sustainable economic development in dammed watersheds. Cross-sectoral cooperation has the potential to yield cross-sectoral benefits.

## 4. Conclusions

As the pace of dam development quickens in low- and middle-income countries, managing the health and livelihood impacts of new and existing infrastructure is crucial to ensuring the sustainability of modern agriculture and economic development in a changing climate. Particularly in schistosomiasis-endemic regions, agricultural innovations involving the restoration of aquatic migratory species, the harvest and transformation of invasive aquatic vegetation, the restoration of environmental flows, and the integration of agriculture and aquaculture hold promise for reducing disease transmission while also improving economic development and food security. These innovations will require careful planning and management as well as cooperation between sectors, but strategically implementing such agricultural innovations can start to turn vicious cycles of poverty and disease into virtuous cycles of health and development.

**Author Contributions:** Conceptualization, A.J.L., D.L.-C. and G.A.D.L.; writing—original draft preparation, A.J.L.; writing—review and editing, A.J.L., D.L.-C., S.H.S., J.R.R. and G.A.D.L.; funding acquisition, D.L.-C., S.H.S., J.R.R. and G.A.D.L. All authors have read and agreed to the published version of the manuscript.

**Funding:** A.J.L. was supported by a James and Nancy Kelso Fellowship through the Stanford Interdisciplinary Graduate Fellowship program. D.L.-C. was partially supported by the National Science Foundation (BCS-1414102). G.A.D.L. and S.H.S. were partially supported by grants from the National Science Foundation (ICER-202438 and DEB-2011179) as well as a grant from the Bill & Melinda Gates Foundation (#OPP1114050). J.R.R. was supported by grants from the National Institutes of Health (R01TW010286-01), the National Science Foundation (DEB-2017785, IOS-1754868) and the Indiana Clinical and Translational Sciences Institute.

**Institutional Review Board Statement:** Not applicable.

**Informed Consent Statement:** Not applicable.

**Data Availability Statement:** Not applicable.

**Conflicts of Interest:** The authors declare no conflict of interest.

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
