# Peer review of "Agricultural Innovations to Reduce the Health Impacts of Dams"

_sustainability, doi:10.3390/su13041869_

Round 1

Reviewer 1 Report

The communication is clear and well written. I only suggest to change the title in:

Agricultural methods to reduce the health impact of dam.

There are two reasons supporting this suggestion:

  1. the Authors describe agricultural already known methods, not advanced processes or innovative methodology
  2. the paper is mainly based on the reduction of the schistosomiasis and not on the equity increase

Author Response

Thank you for the positive feedback and the title suggestion. We have revised the title accordingly, but decided to retain the term “agricultural innovations”, since it used throughout the manuscript and it reflects the theme of the special issue. 

Reviewer 2 Report

I find the discussion of this paper to be interesting, innovative, and policy relevant. I enjoyed reading this paper and appreciate that the authors provided such a thorough discussion for this issue. One major contribution of this paper is that it proposes several potential future directions for researchers to investigate.

Author Response

Thank you for the positive comments. We are very glad you enjoyed reading the paper and found its contributions to be valuable for future research and policy.

Reviewer 3 Report

The paper on “Agricultural innovations to increase equity and reduce the health impact of dams” presents 4 innovations in order to reduce the negative impact of dams on the environment and human. The paper is interdisciplinary, from the area of agriculture, aquaculture, human health and society. It is well suited for Sustainability journal. Presented innovations are related to sustainable development and sustainable agriculture goals. These innovations can be widely used, but they are especially important in the schistosomiasis regions.

These innovations are well described. The discussion is based on comprehensive literature. The paper is written using good scientific language. But the paper is theoretical. It is based on the review of literature and own ideas according innovations. There is no own research on the presented subject. Probably it is enough for the type of paper – “Communication”. But there is no prognosis of long-term effects of these innovations, for example on biodiversity. On the other hand, presented solutions are interesting and can be disseminated and implemented in some countries.

The conclusion section is quite long. I suggest to include it in the discussion and prepare new short conclusion. Small remarks were put in the pdf.

Author Response

Thank you for the positive comments. You’re right that we do not present original research, which is why we submitted this piece as a Communication. However, our insights come from many years of working in rural Africa performing social and ecological research on the impacts of dams on human disease, and the potential solutions (many of our original research works are cited in this piece, see references 16, 18, 29-31, 34-37, 40, 46-47, 55, 72).  We hope that this paper will stimulate future research and policy discussions on how the agricultural sector can contribute to disease control.

We appreciate the remarks in the pdf and have made the suggested revisions. Additionally, the original Conclusion section is now Discussion and we have added a new short Conclusion to follow the Discussion.  

Reviewer 4 Report

Dear Authors, thank you for the opportunity to review this interesting manuscript. The paper is filling a lack of knowledge in the current scientific literature review about the appropriately addressing the livelihood, health and environmental impacts of dams, as well as for the innovative low-cost ways to restore dammed watersheds’ ecological integrity.

The aim and the objectives are clear. In general, I can say that the paper contains new and useful information.

The relevant scientific literature is very well reviewed and discussed.

The paper demonstrates an adequate understanding of the proper knowledge in this specific field of the livelihood, health and environmental impacts of dams, in correct connection with results and suggestions. Some statements in the paper should be supported by more evidence and the relevant references (eg lines 112-113, 117-124, 147-150, 151-153, 155-157, 226-229).

The literature review method is appropriate and findings remarkably interesting.

Results are clearly presented, and the discussion provide a better understanding of them, as well as the conclusions, which are consistent with the development and the findings of the study. The practical implications of the research show an in-depth analysis of the research literature and the objective.

The paper has good quality of communication, in general is well structured and quite easy to read.

I agree with the Authors that addressing the livelihood, health and environmental impacts of dams will become central to improving the sustainability of modern agriculture and economic development in a changing climate. There are many gaps in the current research on this specific field

Author Response

Thank you for the positive comments and helping us identify where we can further support our statements with references. We have added references to all but one of these places. In lines 113-116 of the revised manuscript, we have instead revised the text to make our logical argument clearer.